



Tangjiaxi Landslide and Impulse Wave Analysis in Zhexi Reservoir of
China by Granular Flow Coupling Model
Huang Bolin[1]✉, Yin Yueping[2] Wang Shichang[1], Liu Guangning[1], Tan Jianmin[1]
1. Wuhan Centre of China Geological Survey, Wuhan, China, 430205; 2. China Institute for
Geo-Environment Monitoring, Beijing, China, 100081.
E-mail: bolinhuang@aliyun.com
**Abstract:** Rocky granular flow usually forms after rocky bank slopes are failed and rushes into
rivers at a high velocity, causing impulse wave disasters. Currently, the granular mass/water
coupling study is an important trend in the field of landslide-induced impulse wave. In the paper, a
full coupling numerical model for landslide-induced impulse wave is built based on non-coherent
granular flow equation. In this model, Mih equation for continuous non-coherent granular flow
controls the movement of sliding mass, two-phase flow equation regulates the interaction between
sliding mass and water, Re-Normalisation Group (RNG) turbulence model governs the movement
of water body. Taking Tangjiaxi landslide as an example, which is located at Zhexi Reservoir in
Hunan Province, China, the motion characteristics of Tangjiaxi landslide and the following
impulse wave process were analyzed by the coupling model, and the validity of this model was
checked. On July 16, 2014, rocky blocks debris flow was formed after the failure of Tangjiaxi
landslide, damming Tangjiaxi stream and thus causing an impulse wave disaster with which left
three persons dead and nine persons missing. The full coupling numerical analysis showed that
after the failure of Tangjiaxi rockslide, rocky granular flows impacted the water at the maximum
velocity of about 22.5 m/s, with waves propagating at the maximum celerity of up to 12 m/s. The
deposited topographic modeled is similar to that accumulated in the actual situation. The
maximum run-up calculated is 21.8 m, close to the value of 22.7 m obtained in the field survey. A
series of run-up values in the field survey matches well with the calculated values. Therefore, the
full coupling numerical model built in this study can be used to simulate impulse waves generated
by rocky granular flows.

**Key words:** granular flow; coupling model; Tangjiaxi landslide; impulse wave; landslide dam

**1. Introduction**
Impulse waves are usually generated in reservoirs, rivers, lakes and seas as rock/soil masses
impact water, resulting in huge economic losses and casualties (Wang et al. 1986; Fritz 2001;
Scheffers and Kelletat 2003; Alvarez-Cedrón et al. 2009; Silvia et al. 2011; Huang et al. 2012).
This fact urges people to pay attention to landslide-induced impulse wave which is an
interdisciplinary study related to rock/soil mechanics and fluid mechanics. A large number of
researches have been done on landslide-induced impulse wave with formulae, physical experiment
method and numerical analysis method. The formulae derive from extensive sources, such as
experiment and empirical formulae, with its application scope closely related to sources
(Kamphuis et al. 1970; Ataie-Ashtiani et al. 2008; Wieland et al. 1999; Ursell et al. 1960; Fritz et
al. 2002; Huber and Hager 1997; Heller 2007; Yin and Wang 2008). Due to relatively simple
results after calculation by the formulae, it is hard to have an overall grasp of the





landslide-induced impulse wave disaster (Heller et al. 2009). The scaled physical experiment method can well reproduce or preview the process of how landslide induces impulse waves (Ball 1970; Davidson and Whalin 1974; Muller and Schurter 1993), but it need large data, occupy big space, spend much money, and take a long time (Huang et al.2014). However, the numerical analysis method can help us have a relatively comprehensive analysis of the landslide-induced impulse wave disaster; it has the advantage of precise, economic and reasonable, as well as highly visible results (Heller et al. 2009). Therefore, the numerical analysis method is an important tool in the study of landslide-induced impulse wave (Yuvari-Ramshe and Ataie-Ashtiani, 2016).

In the field of granular mass/water body coupled numerical analysis, three main numerical simulation methods are now used to analyze the landslide-induced impulse wave disaster, i.e. a) single model for landslide-induced impulse wave, b) simplified model for landslide-induced impulse wave, and c) full coupling model for landslide-induced impulse wave (Yuvari-Ramshe and Ataie-Ashtiani, 2016). Their numerical calculation is constructed by the mesh-based methods (finite difference method (FDM), finite element method (FEM), finite volume method (FVM), boundary element methods (BEM), et al.), meshless-based methods (smoothed particle hydrodynamic (SPH), material particle method (MPM), et al.), and particle-based discrete element method (Yuvari-Ramshe and Ataie-Ashtiani, 2016).

In the single simulation method for landslide-induced impulse wave, the phase of landslide movement and granular mass/water body interaction are regarded as the formation of initial impulse wave, and generally the motion of the sliding mass is considered to the motion of a point. Therefore, various kinematic formulas, such as Newton's laws of motion, are applied to calculate the motion of the sliding mass (Heller 2009; Huang et al. 2012, 2016). Then various empirical or experimental formulas of landslide-induced impulse waves are adopted to calculate initial impulse wave caused by the landslide (Walder et al. 2003; Tappin et al. 2008; Watts et al 2003; Ataie-Ashtiani and Malek Mohammadi 2007). With the initial impulse wave as the initial input or boundary condition, the numerical simulation singularly aims at calculating the spread and run-up of impulse waves. This type of numerical simulation models includes TUNAMI, MOST, FUNWAVE, COULWAVE, etc. (Joseph et al. 2003; Rahiman et al. 2007; Tinti et al. 1999; Tappin et al. 2008; Eric 2009). Their accuracy and application scope largely depend on the source models for initial impulse wave. Many scholars (Watts et al. 2003; Ataie-Ashtiani and Malek-Mohammadi 2008; Di Risio et al. 2011; Yin et al. 2015) have studied initial impulse wave models in different range of application and come up with a large number of source models.

The simplified simulation for landslide-induced impulse wave means to simplify landslide motion in calculation. Some landslides are simplified as rigid bodies whose motion is mainly described with Newton's law of motion under gravity, friction, coupled water resistance, etc. (Das et al., 2009; Basu et al., 2009; Huang et al., 2013). For example, Yin et al.(2014) simulated the motion of Qianjiangping landslide as a rigid rotator and coupling calculated the impulse waves. Harbitz et al. (2014) simulated a rockslide with the volume of $5 \times 10^7$ m$^3$ at western Norway Åkerneset fjord as a rigid sliding block. Such simplified methods can reveal the rules of how various dynamic models of a rigid body affect impulse waves (Yin et al., 2015). For some flow-liked slides or debris flow, simple fluids or grains are used to simulate large deformation in the process of the motion of landslide. For instance, Ren et al. (2006) simulated the motion of Xintan landslide by regarding it as some large grains which complies with Newton's laws of motion and the law of conservation of energy. Gabl et al. (2015) used fluid to simulate landslide




occurred at hillsides and the following impulse waves. Abadie et al. (2010) adopted the
multi-phase flow model to simulate landslide-induced impulse waves, as a Newtonian fluid
simulating the landslide. In these researches, simple fluids or grains are used for simplified
simulation and thus the effects of landslide deformation on landslide-induced impulse waves could
be taken into consideration at least partly in calculation.
The full coupling model for landslide-induced impulse wave, is a currently emerging method,
which is booming recently, can have a relatively accurate description of the motion of sliding mass,
interaction with water, and consequent generation, propagation and run-up of impulse waves. As a
simple mathematical motion model has much difficulties in achieving real description of the
motion of landslide, the model mostly used is the complicated rheological model or discrete
element model. In researches so far, models that describe flow-liked landslide or debris flow in
continuous rheological models are Coulomb model, Herschel–Bulkley model, Bagnlod model and
Bingham model (Shakeri Majd and Sanders 2014; Cremonesi et al. 2011; Yuvari-Ramshe and
Ataie-Ashtiani, 2016; Xing et al., 2016). Those that describe avalanche, landslide or debris flow
motions in discontinuous medium models are mainly FEM-DEM model (Morris et al. 2006;
Munjiza 2004; Li et al., 2015) and DEM model (Smilauer et al. 2010; Brennen 2005; Utili et al.
2014). For generation, propagation and run-up of impulse waves, technologies that can finely
depict large deformation free surface, such as VOF or non-hydrostatic models (Yuvari-Ramshe
and Ataie-Ashtiani, 2016) are adopted. Crosta et al. (2013) used an ALE-FEM approach for a
2D/3D simulation of landslide and impulse wave. Glimsdal et al. (2013) developed a model for
submarine landslide and tsunami, the landslide motion was simulated as a deformable viscoplastic
Bingham fluid. Zhao et al. (2015) used 3D DEM-CFD coupling method to simulate the motion of
vajont landslide and the resulting impulse waves. By combined landslide dynamic model and
tsunami model, Sassa (2016) presented an integrated numerical model simulating the complete
evolution of a landslide-induced tsunami, and this model was applied to the 1792
Unzen-Mayuyama mega slide and tsunami disaster analysis.
In the paper, a full coupling model for landslide-induced impulse wave based on non-coherent
granular flow equation is built and then the continuous granular flow Mih (1999) model is
introduced to simulate the process of rocky granular motion after rockslide, and the two-phase
flow model is adopted for interaction coupled calculation. Taking Tangjiaxi rockslide and the
resulting impulse wave as a case, a numerical analysis for the whole process is done to study the
motion of the granular flow, its accumulation process and consequent formation, propagation and
run-up of impulse waves. Meanwhile, the validity of the full coupling model for landslide-induced
impulse is checked.
**2. Theory and Methodology**
Rockslides can be characterized by a rapid evolution, up to a possible transition into a rock
avalanche, which can be associated with an almost instantaneous collapse and spreading (Utili et
al. 2014). After rocky slopes fail, high concentration and non-coherent rocky granular motion
follow. A large amount of non-coherent coarse solid grains as well as relatively few fine grains are
densely distributed in the granular flows. They flow, deposit or erode along their motion routes,
which spread very long in distance generally (Crosta et al. 2001). Such flowing characteristics of
motion can be described through both the continuous rheological model and the discontinuous
model. The discontinuous model features natural intuitive similarity when used to study the
motion of non-coherent granular flows. For the discontinuous method, grains are generally





simplified to be sphere. These grains can interact with each other through well-defined
microscopic contact models (Hertz 1882; Zhang and Whiten 1996; Johnson 1985) and with the
fluid (e.g. water or air) by empirical correlations of fluid and solid interaction models. However,
the discontinuous method means a large challenge for individual researchers. That is because even
for a small rockslide, the simulation will require numerous cells and huge computational resources,
hard to be processed by personal computers (Utili and Crosta 2011). Whereas the model based on
continuous granular flow is free from this problem.
The continuous granular flow model is built by using viscous fluid. In this field, high
concentration granular flow was studied by Bagnold (1954), Savage (1978), Hanes and Inman
(1985), Wang and Campbell (1992), Iverson (1997) and Mih (1999).

**2.1 Governing equations of granular flow**

Landslide rheology describes landslide motions with shear stress ($\tau$) or shear rate (Pudasaini 2011).
Shear stress of granular flow is generally far larger than the cohesive shear stress of fluids that
carry a small amount of grains. Shear stress in high concentration non-cohesive granular flow ($\tau_g$)
consists of: (1) Impact among solid particles ($\tau_i$); (2) Additional viscous shear stress due to the
presence of solid particles ($\tau_v$); and (3) Shear stress in the fluid ($\tau_f$) (Mih 1999). It becomes
negligible in solid-gas flow when the dynamic viscosity of the gas is small. At high concentrations
the principal contribution to the shear stress arises from impact forces (i.e., collision) among
grains. Secondly, in general, smaller contribution arises from the distributed solid affecting the
fluid. Bagnold (1954) performed shear cell experiments with different approaches and showed that
an equation for cohesionless materials describes the relationship between bulk intergranular
normal and shear stresses even in collision-dominated flows.
Extensive work, beginning with the 1954 work of Bagnold (1954) has been summarized and
further extended to a larger range of experimental conditions by Mih (1999).The equation for
shear stress of Mih (1999) granular flow is as follows:

$$\tau_g = \tau_v + \tau_i = 7.8\mu \frac{\lambda^2}{1+\lambda} \frac{du}{dy} + \rho_s \frac{0.015}{1+0.5\,\rho/\rho_g} \frac{1+e}{(1-e)^{0.5}} (\lambda D \frac{du}{dy})^2 \qquad (1)$$

Here: $\mu$ and $\rho$ are the continuous fluid viscosity and density between granular (e.g. air or
water), $\rho_g$ is the granular density, $e$ is the coefficient of restitution associated with grain impacts, $d$
is the grain diameter, and $d$ is a function of the maximum solid volume fraction. Physically,
$\lambda = d / S$ where $S$ is defined as the average distance between grain centers, $du/dy$ is the mean
velocity gradient of the granular mixture.
The equation contains fluid viscous and impact coefficients. The fluid viscous coefficient is a
constant. The impact coefficient has been correlated to the properties of the solid and fluid. The
equation agrees reasonably well with several sets of experiments by different investigators which
cover a wide range of granular flows (Mih, 1999).

**2.2 Granular flow/fluid interaction**

The granular flow is treated as incompressible fluid when applied with the shear stress equation of
Mih (1999). The coupling model of granular flow and water adopts two phase model with two
incompressible fluids having different densities. Supposing the water has density $\rho_1$, the granular
flow has density $\rho_2$. The volume fractions of the water making up a mixture is denoted by $f$, and



the volume fractions of the granular is denoted by 1-$f$. The momentum balance for the continuous
phase of water is

$$\frac{\partial u_1}{\partial t} + u_1 \cdot \nabla u_1 = -\frac{1}{\rho_1}\nabla P + F + \frac{K}{f\rho_1}u_r \qquad (2)$$

While for the dispersed phase of granular flow, it is

$$\frac{\partial u_2}{\partial t} + u_2 \cdot \nabla u_2 = -\frac{1}{\rho_2}\nabla P + F - \frac{K}{(1-f)\rho_2}u_r \qquad (3)$$

Where:
$u_1$ and $u_2$ represent the velocities of the continuous and dispersed phases, respectively; F is the
body force; P is the pressure; K is a drag coefficient that relates to the interaction of the two
phases; $u_r$ is the relative velocity difference between the dispersed and continuous phases:
$u_r = u_2 - u_1$          (4)
The volume-weighted average velocity $\overline{u}$ of a mixture is Eq. (5).
$\overline{u} = fu_1 + (1-f)u_2$          (5)
The volume-weighted average velocity momentum conservation equation is Eq. (6).
$\nabla \cdot \overline{u} = 0$          (6)
The drag per unit volume (K) is calculated by Eq. (7).

$$K = \frac{1}{2}A_2\rho_1\left(C_D U + 12\frac{\mu_1}{\rho_1 R_2}\right) \qquad (7)$$

Where:
$A_2$ is the cross sectional area per unit volume of the dispersed phase;
$\rho_1$ and $\mu_1$ are the water density and dynamic viscosity;
$C_D$ is the user-specified drag coefficient. It is a dimensionless quantity and is 0.5 for spheres.
$R_2$ is the average particle size of the granular.

**2.3 Governing equations of fluid flow**
RNG k-ε model is used to calculate the fluid motion when the granular flow into the water. The
RNG model applies statistical methods to the derivation of the average equations for turbulence
quantities, such as turbulent kinetic energy and its dissipation rate. The RNG model uses equations
similar to the ones for the k-ε model. However, equation constants are derived explicitly in the
RNG model, and it takes turbulent vortex into account. Generally, the RNG model has a wider
applicability than the standard k-ε model. The transport equation for $K_T$ includes the convection
and diffusion of the turbulent kinetic energy, the production of turbulent kinetic energy due to
shearing and buoyancy effects, diffusion, and dissipation due to viscous losses within the turbulent
eddies (Yakhot and Orszag 1986; Yakhot and Smith 1992). The transport equation for $K_T$ is:





$$\frac{\partial k_T}{\partial t} + \frac{1}{V_F}\left\{ uA_x \frac{\partial k_T}{\partial x} + vA_y \frac{\partial k_T}{\partial y} + wA_z \frac{\partial k_T}{\partial z}\right\} = P_T + G_T + Diff_{k_T} - \varepsilon_T \qquad (8)$$

An additional transport equation is solved for the turbulent dissipation, $\varepsilon_T$:

$$\frac{\partial \varepsilon_T}{\partial t} + \frac{1}{V_F}\left\{ uA_x \frac{\partial k_T}{\partial x} + vA_y R \frac{\partial k_T}{\partial y} + wA_z \frac{\partial k_T}{\partial z}\right\} = \frac{CDIS1 \cdot \varepsilon_T}{k_T}(P_T + CDIS3 \cdot G_T) + Diff_\varepsilon - CDIS2 \frac{\varepsilon_T^2}{k_T}$$

4      (9)

In the RNG turbulence transport models, the kinematic turbulent viscosity $V_T$ is computed from

$$v_T = CNU \frac{k_T^2}{\varepsilon_T}$$

The diffusion of dissipation, $Diff_\varepsilon$ is:

$$Diff_\varepsilon = \frac{1}{V_F}\left\{ \frac{\partial}{\partial x}(v_\varepsilon A_x \frac{\partial \varepsilon_T}{\partial x}) + R\frac{\partial}{\partial y}(v_\varepsilon A_y R \frac{\partial \varepsilon_T}{\partial y}) + \frac{\partial}{\partial z}(v_\varepsilon A_z \frac{\partial \varepsilon_T}{\partial z}) + \xi \frac{v_\varepsilon A_x \varepsilon_T}{x}\right\} \qquad (10)$$

Where $k_T$ is the turbulent kinetic energy, $V_F$ is the fractional volume open to flow, $A_x$ is the
fractional area open to flow in the x direction, $A_y$ and $A_z$ are similar area fractions for the flow in
the y and z directions, respectively. $P_T$ is the turbulent kinetic energy production term, $G_T$ is the
buoyancy production term, $\varepsilon_T$ is the turbulence dissipation term. In the RNG model, CDIS1,
CDIS3, and CNU are dimensionless user-adjustable parameters that have 1.42, 0.2 and 0.085
defaults. CDIS2 is computed from the turbulent kinetic energy ($K_T$) and turbulent production ($P_T$)
terms (Yakhot and Orszag 1986; Yakhot and Smith 1992).
In particular, the RNG model is known to describe low intensity turbulence flows and flows
having strong shear regions more accurately. The RNG model selected has already been
successfully used to simulate impulse wave generated by landslides (Serrano-Pacheco et al. 2009;
Basu et al. 2009; Das et al. 2009; Choi et al. 2007).
**3. Case Study**
A full coupling numerical analysis model for landslide-induced impulse wave is built based on
coupled control equations. The model can stimulate the landslide motion of non-coherent granular
flow and the generation, propagation and run-up process of impulse waves. The case of Tangjiaxi
landslide in Zhexi Reservoir, Hunan, China, is taken as an example, the whole process of the
landslide and impulse wave induced are analyze, as well as the validity of numerical model.
**3.1 Overview of Tangjiaxi landslide and impulse wave**
At 7 AM on July 16, Tangjiaxi landslide occurred on the left bank of Tangjiaxi Stream, a tributary
of Zhexi Reservoir. The impulse wave induced by Tangjiaxi landslide destroyed resident living
area nearby. The landslide is 700 m far from the mainstream of Chanxi stream (tributary of Zi
River), and 10.6 km and 11.2 km away from Tangyanguang landslide site and Zhexi Dam along
the watercourse, respectively (Fig. 1). Zhexi Dam is located in midstream of Zi River in Anhua
County, Yiyang City, Hunan Province, China, and 15 km away from the seat of Anhua County.
Zhexi Hydroelectric Station, which began to impound in February 1961, is a large hydroelectric
station. Tangyangguang landslide occurred on March 6, 1961. It is the first impulse wave disaster




generated by landslide since the founding of the People's Republic of China. The huge wave
generated by Tangyangguang landslide overtopped Zhexi Dam and killed 64 persons (Du 1988).
The impulse wave disaster generated by landslide happened again in this reservoir, which drew
much attention.

Fig. 1 The location of Tangjiaxi landslide in the Zhexi reservoir, Hunan Province, China

The landform of Tangjiaxi stream valley belongs to the type of medium gorge. The elevation of
the highest mountain in this valley is 650 m, while the bottom elevation is 140-170 m generally.
The overall flow direction of Tangjiaxi Stream is 245°, with a large gradient of about 1 km long.
When water level elevation is 169.5 m, the stream is 2-100 m wide and 2-30 m deep. The original
slope at valley bottom is about 25°~30°, and that at altitude above 200 m was 35°-45°. Generally,
eluvial and diluvial deposit of 2-5 m thick was developed in the slope of the valley, with lush
vegetable covered.
The rain continued for almost half a month from late June to early July in 2014. The daily rainfall
was 98.5 mm around July 4. The Zhexi Reservoir was hit by rainstorm on July 13 again. The
rainfall reached 102.5 mm on July 15 and seriously 239 mm on July 16 (Fig. 2). Rainfall increased
the weight of sliding mass, formed greater underground water dynamic pressure, and decreased
anti-sliding strength (Thomas 2003; Wang et al., 2004). Persistent rainfalls and heavy rainstorm
directly triggered the failure of the landslide.

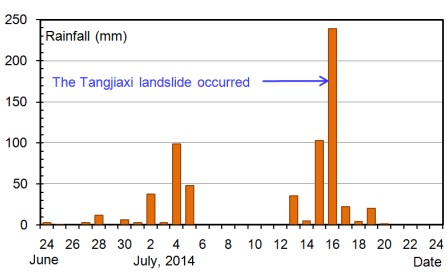
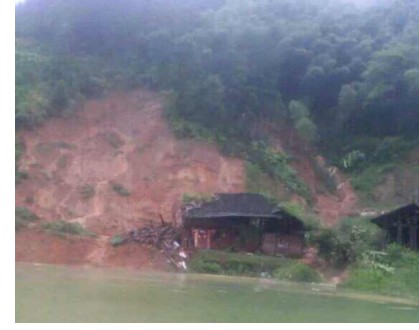





Fig. 2 Precipitation data monitored in Sifang village, 3.6 km from the landslide.

Fig. 3 Photo of first slide, taken by some local villager on July 16, 7 AM.

According to the description of many local survivors, the first slide occurred around 7 AM on
July 16. Fig. 3 shows the scene of the first slide. Starting from the toe of the slope, the first slide
was shallow soil slide which destroyed one of the three houses on the sliding mass. There was a
short quiet period after the first slide. About 10:20 AM, rock blocks rolled down from the top of
the slope and the global slide started. As soon as the landslide mass started to run out, rocks broke,
crashed and rushed rumbly down to the slope foot, and houses were buried quickly. The mass
impacted on Tangjiaxi stream at a high speed and induced huge waves, and the still water level
was 169.5 m above sea level (asl.).
As shown in Fig. 4, the morphology of landslide scar was triangular in shape. The crown
elevation of the landslide was about 315 m and the elevation of the outlet was about 155 m. The
height difference was 160 m. At 26 m above the water surface, the landslide was 95 m wide, and
at 56 m above the water surface, the landslide width reached 80 m. Much closer to the crown, the
width of the landslide was smaller. The landslide was 15 m thick on average, with a total volume
of 160,000 $m^3$, and main sliding direction was 320°.

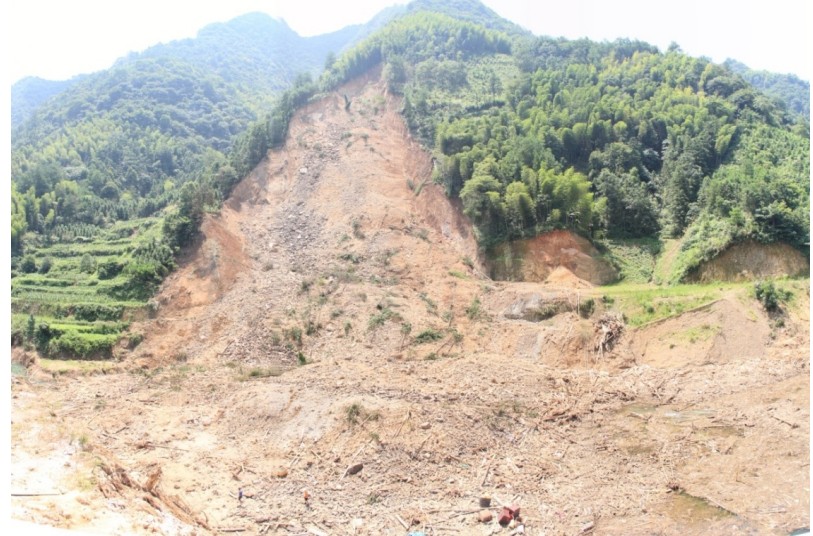

Fig. 4 The scene of Tangjiaxi landslide, taken on July 23, 2014, when the water level was 167 m asl. The river was full of wood and debris, which were the destroyed building materials.

The underlying bedrock of Tangjiaxi Slope is Nantuo Formation ($Z_n$) and Guanyintian
Formation of Sinian ($Z_g$) according to drilling reconnaissance and field survey. The lithology is
grey-green till conglomerate and red metamorphosed quartz sandstone. The dip of schistosity of
the rock mass is 300°-310° with the dip angle of 30°-40°. Two groups of faults with high dip angle
are developed under the slope, which strike direction is nearly parallel to the valley. The fault belt
is mylonite mainly (Fig. 5). Influenced by the fault, fissures are developed and there are mainly
two groups of the structure planes: 1. fissures with a dip of 20°-30° and a dip angle of 60°-70°; 2.



fissures with a dip of 300°-320° and a dip angle of 65°-70°. Red or brown clay can be seen in
some fissures. Two groups of structural planes and schistosity intersected mutually cataclasite
structure rock mass were formed in Tangjiaxi slope.

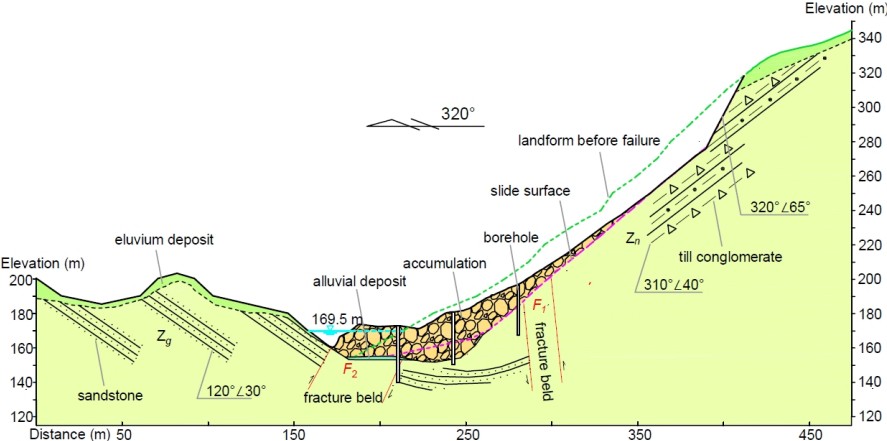

Fig. 5 Geological engineering section of Tangjiaxi landslide

After the landslide failed, cataclasite structure rock mass disintegrated quickly. The
accumulation of sliding mass was mainly composed of rock blocks of different sizes. Medium and
large rock blocks were mainly in the lower-middle part, with the maximum length of rock blocks
of about 2.5 m. Rock blocks in the accumulation, having the shape of sharply angular with an
average diameter of 30-40 cm, overhead stacking (Fig. 6). The few gravelly soils on the
accumulation site were mainly distributed on the flanks of the landslide and at the front edge of
accumulation fan. These soils were mainly derived from weathered layer and eluvial deposit of the
original slope.

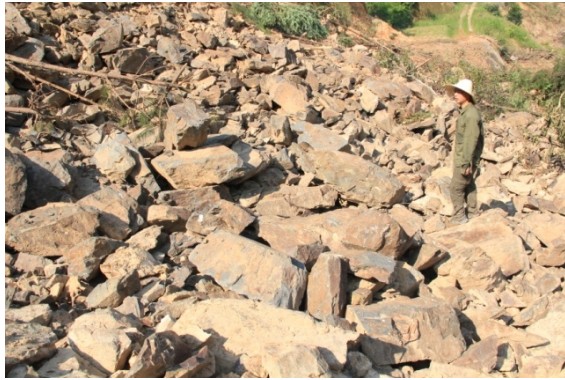

Fig. 6 accumulated blocks after Tangjiaxi landslide failure, taken in July 23, 2014.

Part of the sliding mass was accumulated in the watercourse and some stayed on the slope. The
landslide dam raised the river bed and halted part of upstream water to form a small landslide lake.





The landslide dam was high in downstream and low in upstream, with bulge in the middle. Two
terraces were formed on the vertical section. The dip angle of the deposits on the terrace was about
33°. The first slope terrace had an average elevation of about 180 m, 38 m long and 77 m wide,
with a gradient of about 10°, while the second terrace had an average elevation of about 172.5 m,
75 m long and 98 m wide, with an average gradient of about 5-10°. The bulge was in the second
terrace, with the top point of the elevation a.s.l. of about 175.5 m. The river was broken by the
second terrace of the landslide, which could be seen obviously in Fig. 7.

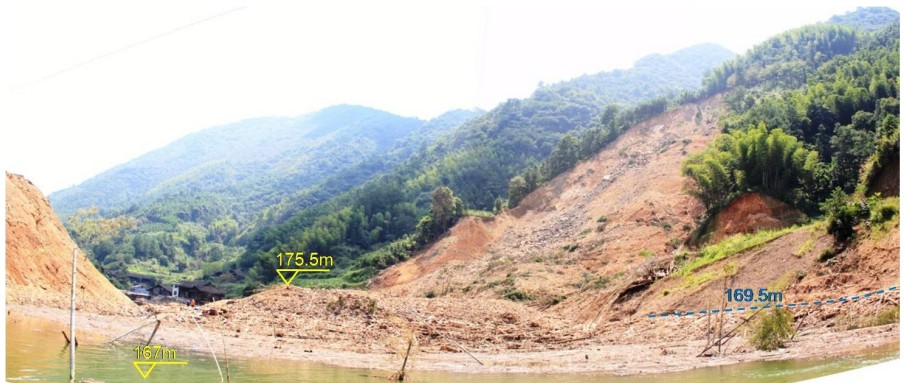

Fig. 7 Profile photo of Tangjiaxi Landslide, taken on July 23, 2014, when the water level is 167 m asl.

Witnesses described that it took only several seconds for the landslide to slide into the water
and form the landslide dam. Calculated by 10 seconds for the sliding duration time, the landslide
barycenter is about 70 m above still water surface and the sliding distance is about 120 m. It is
estimated roughly that the biggest impact speed is about 24 m/s according to Newton's laws of
motion. Huge impulse waves were triggered by the high speed landslide. The impulse wave
attacked the opposite bank, razed 6 houses to the ground, and cut trees to the root (Fig. 8 A). And
then, the impulse wave flowed both upstream and downstream. The high-speed wave destroyed all
houses (Fig. 8 B&D) and trees (Fig. 8 C) it met. 9 houses were destroyed in this tsunamis event, 8
houses damaged and 121 persons of 17 families affected. The impulse wave caused three deaths,
nine people missing, and eleven people wounded, six of which were badly hurt. Fortunately,
owners of 5 destroyed houses went out for work and did not stay in the houses. Otherwise, the
casualties would be more serious.
Though the watercourse in the landslide zone was only about 10m in average, the limited water
gained great energy from the rock blocks granular mass at a high speed and formed huge impulse
waves. As shown in the field survey, the maximum run-up was 22.7 m occurred in the opposite
bank of the landslide; the upstream maximum run-up was 19.5 m occurred in a gully about 100 m
upstream. At the downstream, with the increase of distance from the source of impulse wave, the
run-up decayed. The maximum run-up at river mouth where Tangjiaxi stream flowed into the
Chanxi stream was 1.8 m(Fig.8). As the Tangjiaxi Stream flowed into the Chanxi Stream nearly
vertically, the water surface suddenly became very wide, impulse wave decayed rapidly and no
sign of impulse wave was seen on either bank of Chanxi stream.





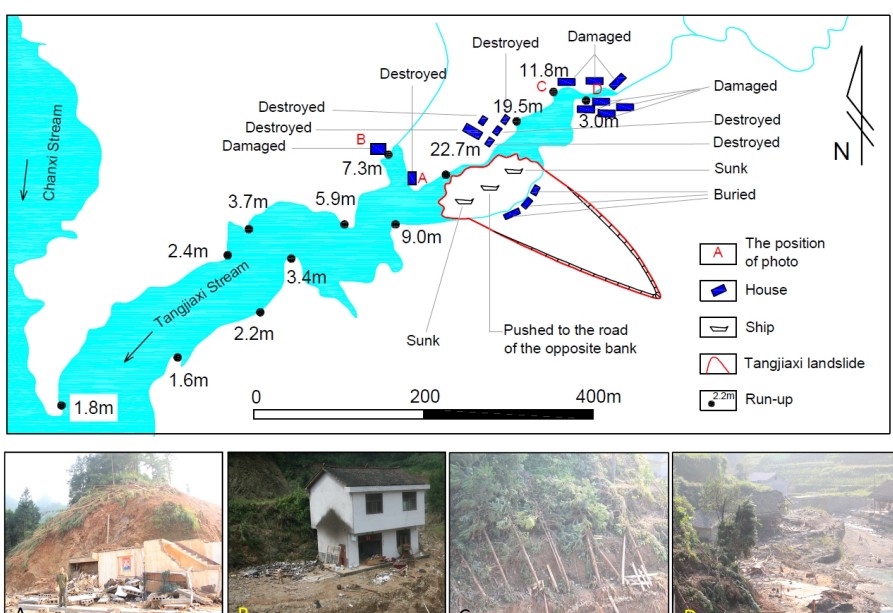

Fig. 8 The plot of run-up of the impulse wave generated by Tangjiaxi landslide, and the photos describe the scene of houses and trees damaged where marked by A, B, C and D in the upper map.

**3.2 The granular flow coupling model**
The full coupling numerical model for Tangjiaxi landslide-induced impulse wave is built based on
the landforms of the valley where Tangjiaxi landslide occurred. The model is 792 m long and 684
m wide. The model area covers the valley source of Tangjiaxi stream at the tail of Zhexi Reservoir,
with the lowest elevation of 140.0 m and the maximum mountain elevation of 740.2 m (Fig. 9).
The digital elevation model of Tangjiaxi sliding mass is plotted based on the drilling survey and
the topographic maps before and after the landslide, with a volume of about 158,000 m$^3$. Tangjiaxi
landslide model is set to be a granular flow model. As Tangjiaxi landslide failed under the
condition of persistent rainstorm, the gaps between grains were basically filled with rainwater.
Thus, the fluid in Tangjiaxi landslide granular flow gaps was water. During the process of
Tangjiaxi landslide motion, there were two distinct phases for the motion of rocky grains: start-up
and moving phase and impact-stop phase in sequence. Impact in the first phase mainly occurred
among grains and that in the second phase mainly between leading grains and the opposite bank.
Therefore, two elastic restitution coefficients were adopted, and 0 was taken in the second phase
when the leading granular flow impact the bank. Parameters required for granular flow motion
calculation are as shown in Table 1. The parameters of density, average diameter and initial
porosity of rock grains were determined through field survey and laboratory tests. Tangjiaxi
sliding mass was in stationary initially and started moving under gravity. The granular flow moved
and coupled with water after exposure to the river water.
Table 1 Main Parameters for Mih Equation Calculation

| Parameter | Value | Parameter | Value |
|---|---|---|---|
| Fluid density | 1000 | Grain restitution coefficient | 0.2/0 |





| Fluid viscosity | 0.001 | Average grain diameter | 0.4 |
|---|---|---|---|
| Grain density | 2640 | Global vent coefficient | 0.001 |

The water surface elevation in the model is 169.5 m asl., and the still water surface is the initial
condition. Xmin surface is the zero flow boundary to ensure a constant water volume of Tangjiaxi
stream. Zmax (water surface) is zero pressure boundary or free surface. Zmin surface, Xmax
surface, Ymin surface and Ymax surface are all solid wall surfaces which is far away from the
valley, so they are also zero flow boundaries. With the finite element/volume method adopted,
there are 13,001,472 units in total in grid of 2 m × 2 m × 2 m. The simulation calculation of the
numerical model lasts 30 s, After 6 s, the model come into the phase II as the leading granular
flow impact the bank based on trial calculation.

Fig. 9 Numerical model for Tangjiaxi landslide-induced impulse waves. Red points refer to the velocity monitoring points of the sliding mass motion and blue ones refer to the process monitoring points for water level.

**3.3 Results**
The coupled results were analyzed in the following aspects: the motion process of the sliding mass
and the process of impulse wave. And the model's validity was also checked through comparison
with the field survey results.
**3.3.1 Landslide movement process**
Upon the start of the model analysis, the sliding mass started to move. From the depth-averaged
velocity curves at different elevation points in the sliding mass, it can be seen that the time that
different parts of the sliding mass took to reach the maximum velocity varied. Generally the parts
of sliding mass reached the maximum velocity before the sliding mass impacted the opposite
valley (the 6th second). The maximum sliding velocity of the area at the rear edge (V0) was about
16.6 m/s; that at the middle of the sliding slope (V2) was about 30.9 m/s, possibly the maximum
motion velocity of the sliding slope. V3 point located at the riverside with an elevation of 169.5 m,
V3's velocity approximated to the speed at which the sliding mass impacted water, up to 22.5 m/s
(Fig. 10). The value was equivalent to the maximum impact velocity estimated in field, which is





24 m/s. After the sliding mass impacted the opposite valley, the motion velocity of different parts
of the sliding mass dropped sharply; when it went to about 10 s, the value at the middle and lower
parts of the sliding mass was generally lower than 1 m/s, and that at the upper part was lower than
3 m/s. After 19 s, the velocity of the sliding mass was lower than 1 m/s in overall.

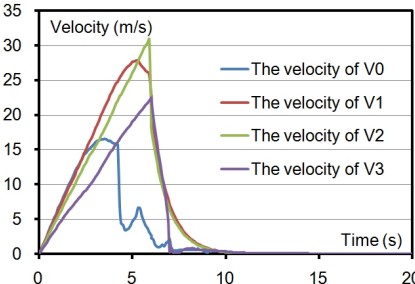

Fig. 10 Depth-averaged velocity process plot of monitoring points in the sliding mass. See Fig. 9
for positions of VO--V1.

Observed from the landslide configuration at different time, the motion of the sliding granular
flow on land is generally within the scope of the sliding mass. After t=4.0 s, the sliding mass
started to occupy the watercourse and extended to the upstream and the downstream, forming a
fan shape (Fig. 11). It can be seen from the comparison with the final plane shape of the
watercourse that numerical simulation results show a more ideal fan-shaped accumulation
(Mohammed and Fritz 2005), and that the landslide dam shape formed in the numerical simulation
differed from the actual situation (Fig. 12). This was possibly attributed to the presumption in the
numerical model, i.e., the solid gains are ideally spherical, with a similar grain size.

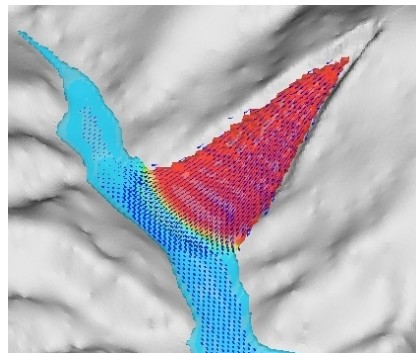

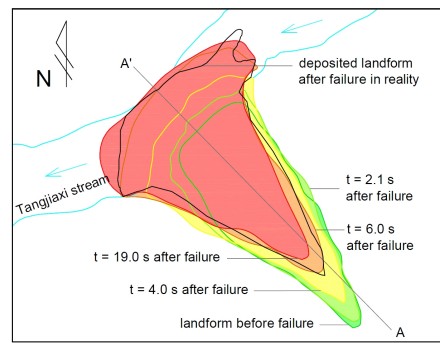

Fig. 11 Instantaneous state of Tangjiaxi
landslide and river surface at t=4.0 s. In the
figure, the red area is Tangjiaxi sliding mass,
the cyan one is water, and the blue arrow is the
motion direction of unit mass points.

Fig. 12 Changes of plane shape after Tangjiaxi
landslide failure

From the A-A' section dynamic process of the landslide in Fig. 13, we can see that as the time
went, solid grains of the sliding mass gradually moved to the valley and accumulated. At t=2.1 s,





substances in the sliding mass slid to the river bed. Substances with an elevation of over 200 m
moved at high velocity, so sliding mass in the area started to get thinning. After 2.1 s, the sliding
mass started to occupy the river bed in a large scale. At t=4.0 s, a small accumulated platform
appeared in its early form in the valley, and kept moving to the opposite. At t=6.0 s, the leading of
the sliding mass impacted the bank slope of the valley, when the landslide formed a large sliding
dam in the valley and almost dammed the watercourse. At t=19.2 s, the landslide configuration
was similar to that at t=6.0 s, and it almost kept unchanged from then on, thus forming a landslide
dam with an average elevation of about 171 m. The actual average elevation of the landslide dam
formed was about 172.5 m. From the section landform after the landslide deposited, we can see
that the actual landform after landslide had an obvious two-step platform while the simulated
result was only large one-step landslide platform, but their surface lines were similar.

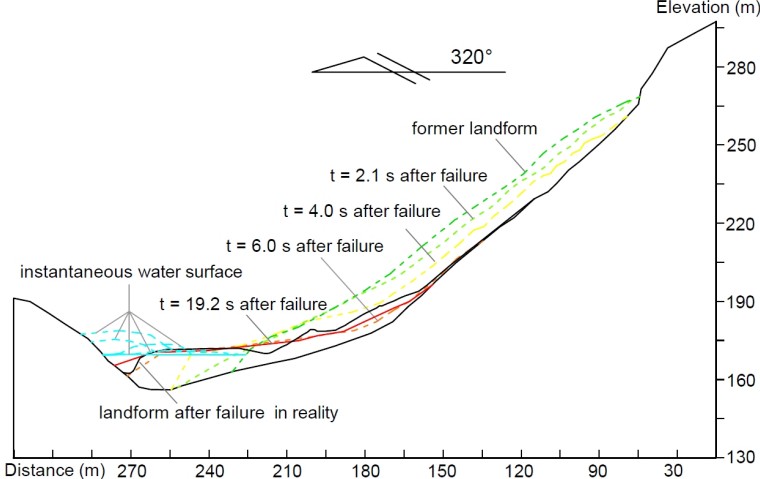

Fig. 13 A-A' Section form after Tangjiaxi landslide failure

**3.2 Process of impulse waves**
The motion results of Tangjiaxi landslide simulated by the granular flow model don't show
significant differences from that in the field survey, basically reflecting the real motion process
and characteristics of the landslide. Huge impulse wave was induced in stream due to the motion
of granular flow.





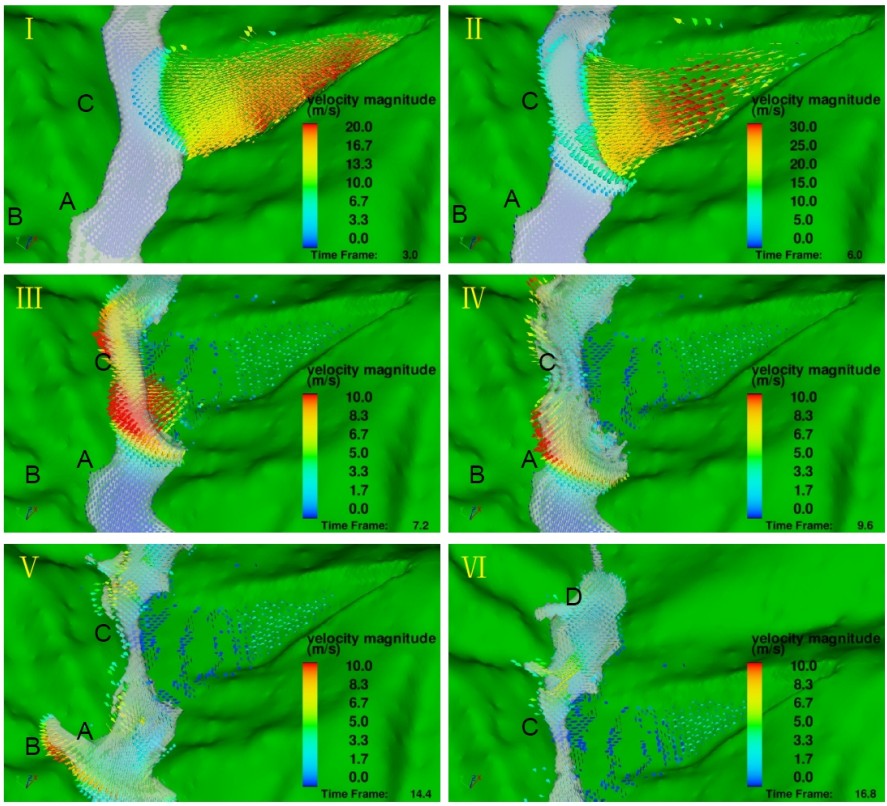

Fig. 14 Transient condition of river water and the vector diagram of mass. The arrow indicates the direction of movement, the color indicates the magnitudes shown in legend.

After the sliding mass occupied the watercourse, it pushed and supported the river water to
move outwards and upwards in an arc shape (Fig. 13 and I in Fig. 14), similar to the forming of
the impulse wave induced by Qianjiangping landslide. At t=6.0 s, an arc-shaped water wall formed
on the river surface, about 10 m high and with the maximum water velocity of about 12.0 m/s,
impacting the opposite and the upstream and the downstream (II in Fig. 14). The residential area
in Area C was impacted firstly at the maximum impact velocity of 11.5 m/s (III in Fig. 14),
resulting in a maximum run-up of 16.5 m in the area. At t=9.6 s, water reached to the ridge near A,
with the maximum traveling velocity of 12.1 m/s (IV in Fig. 14). At t=11.1 s, water flowed over
the ridge and impacted to houses in A, with the maximum velocity of 11.6 m/s. At t=14.4 s,
impulse waves started to impact houses in B, with the maximum velocity of about 7.0 m/s (V in
Fig. 14). After 16.3 s, impulse waves spreading to the upstream reached the residential area in D,
with the maximum water flow impact velocity dropping to 3.8 m/s (VI in Fig. 14). Based on
calculation, the duration from the time the sliding mass started to the time impulse waves attacked
the houses was about 20 s. The impulse waves attacked at high velocity and caused serious house
damages and heavy casualties in the area.
We can also see from Fig. 2 that as Tangjiaxi valley was narrow, the phases of generation,
propagation and run-up of the impulse wave were hard to distinguish at the reach where the





landslide slid into water, so it was not a typical process of impulse waves. As shown in the water
level process line of various points in Tangjiaxi river surface (Fig. 15), there was only one large
peak for the impulse waves in the landslide, especially typical at the reach where the landslide slid
into water (H3 in Fig. 15). Since the upstream of the landslide was quickly dammed after impulse
waves arrived, water reaching the upstream failed to flow smoothly and therefore formed
temporary upsurge in the upstream (Wang et al. 1986). The maximum upsurge in the upstream was
up to 172.5 m (H2 in Fig. 15) and the upstream water level remained about 171. 6 m at 30 s. After
a relatively large impulse wave, wave amplitude fluctuation in the landslide downstream
watercourse attenuated (H4 in Fig. 15).

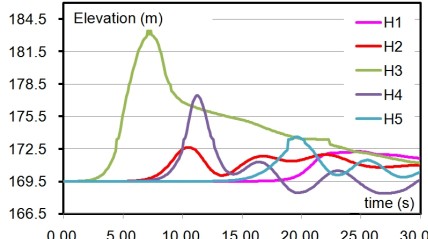

Fig. 15 Hydro Process Line of Various Points in Watercourse. See Fig. 9 for locations of H1--H5.

During the generation of this atypical landslide-induced impulse wave, it was hard to determine
the maximum height of the first wave in the watercourse. The maximum propagating height of the
wave in the peripheral watercourse of the landslide zone was about 8.0 m, located at the
downstream of the landslide. The maximum run-up of the landslide was calculated to be 21.8 m at
the opposite bank of the landslide; the run-up of this point in the field survey was 22.7 m. The
slope at the opposite bank of the landslide was directly impacted by the impulse wave, with
relatively higher run-up. In overall, the run-up was higher in the area where the landslide slid into
water and gradually decreased in the periphery with the increase of distance. Table 2 shew the
run-up at the bank surveyed in the field and corresponding calculated values. The correlation
coefficient ($R^2$) of these two sets of data was 0.98, with an average error of 11%, indicating that
calculated results had high goodness of fit with actual survey results, so the numerical model for
landslide-induced impulse wave is reasonable and valid.
Table 2 Run-up Obtained in the Field Survey and Corresponding Values Calculated

| North | Investigation | 2.4 | 3.7 | 5.9 | 7.3 | 22.7 | 19.5 | 11.8 |
|---|---|---|---|---|---|---|---|---|
| Run-up (m) | Calculation | 3.3 | 3.6 | 6.5 | 7.0 | 21.8 | 17.3 | 12.1 |
| South | Investigation | 2.2 | 3.4 | 9.0 | 3.0 | | | |
| Run-up (m) | Calculation | 3.2 | 4.1 | 9.2 | 3.7 | | | |


**4. Conclusion**
In the paper, a full coupling numerical model for landslide-induced impulse wave was built,
non-coherent granular flow Mih model was used to simulate the dynamic characteristic of
Tangjiaxi rockslide, and the two-phase flow model and RNG model were used to simulate the
impulse waves while the granular flow impacted water.
Tangjiaxi rocky granular flow slid into the watercourse and then moved to the upstream and the
downstream, forming a fan shape, and deposited to be a landslide dam in the valley, damming the





watercourse. The sliding mass impacted water at the maximum velocity of 22.5 m/s, and at the moment the maximum celerity of wave was 12.1 m/s. It was an atypical impulse wave at the reach where the landslide slid into water, where the phases of generation, propagation and run-up of the impulse wave wave were hard to distinguish. The impulse wave induced by the landslide directly attacked the opposite residential area, with the maximum run-up of 21.8 m as calculated. Landslide dam formed hindered the downward flowing of water in the upstream, causing temporary upsurge.

Landslide dam configuration and impulse wave run-up calculated were well fit with the actual survey results. Therefore, the coupling model based on non-coherent Mih granular flow performed well in the whole-process analysis of Tangjiaxi landslide induced impulse wave. The framework of this coupling numerical model deserves more attention and further improvement.

**Acknowledge:**

This work was supported by National Natural Science Foundation of China (project ID: 41372321) and National Science and Technology Support (ID: 2012BAK10B01). Also, the authors would like to thank Mr. Xie from Tangjiaxi village who provide his photos and other useful information to us.

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
