# Peer review of "Analysis of the Tangjiaxi Landslide-generated Waves in Zhexi Reservoir,"

_Natural Hazards and Earth System Sciences, 2016_

## Referee Comment (RC1) · Anonymous Referee #1 · 27 Jan 2017

Landslide-induced impulse wave is common geo-disaster in reservoir, river and sea. It has been done with formula, physical experiment method and numerical analysis method. This manuscript presents a case study of impulse wave analysis by granular flow coupling model. Further discussions on some important issues are necessary, as specified below in the comments.

1. Landslide moved velocity is very important to impulse wave, and the parameters for numerical simulation dominate the results right or not. What are the parameters for calculating the landslide moving velocity, and how to determine them? What about the parameters for fluid?

2. The run-up of the wave is shown in Fig.8. Please introduce the methods to obtain

the heights of wave and the wave kind.

3. The landslide dam shape formed in the numerical simulation is different from the actual situation (Page 13, Line 12) in Fig. 12 and Fig.13. Is it the main reason that the spherical solid gains with similar grain size? How about the influence of soil or rock parameters? Or the parameters in the manuscript were not reasonable?

4. The results in Table 2 have some differences between investigation and calculation. What are the reasons? Further discussions should be added to the results in the manuscript.

5. There are some spell mistakes in the manuscript, i.e. in Fig.1 the Tangyangguang landslide or Tangyanguan landslide.
* * *

---

## Referee Comment (RC2) · Anonymous Referee #2 · 1 Apr 2017

Dear Prof. Chang,

Please attached find my comments on the manuscript entitled as "Tangjiaxi Landslide and Impulse Wave Analysis in Zhexi Reservoir of China by Granular Flow Coupling Model" as a reviewer. I hope you find my comments helpful and constructive. Please feel free to contact me in case you need any further information or you have any question regarding my comments.

Best regards,

Saeedeh Yavari Ramsheh Research Associate Dept. of Civil Eng., Sharif Univ. of Tech., Tehran, Iran Email: yavari@mehr.sharif.edu, saeedeh.yavari@gmail.com

[Figure]

Please also note the supplement to this comment:
http://www.nat-hazards-earth-syst-sci-discuss.net/nhess-2016-332/nhess-2016-332-RC2-supplement.pdf

———————————————
[Figure]

**Supplement:**

Prof. Kang-tsung Chang
Editor
Natural Hazards and Earth System Sciences

31 March 2017

Ms. Ref. No.: nhess-2016-332

Dear Prof. Chang,

In this letter you can find my comments on the manuscript entitled as "Tangjiaxi Landslide and Impulse Wave Analysis in Zhexi Reservoir of China by Granular Flow Coupling Model" as a reviewer. I hope you find my comments helpful and constructive. Please feel free to contact me in case you need any further information or you have any question regarding my comments.

Best regards

This manuscript introduces a granular flow coupling model to study landslide-generated water waves. Landslide motion is described based on Mih (1999)'s equation. The sliding mass/water interactions are modeled using a coupled system of model equations for a two-phase flow. Finally, the water wave characteristics are simulated based on a RNG k-ε model. The model is validated and applied for the Tangjiaxi landslide event in Zhexi reservoir, Hunan, China. I found this paper interesting. However, some parts of the manuscript are expressed so poorly worded that makes them ambiguous and confusing for the readers. A narrative English speaking person should read through the text. The paper can be considered for publication in NHESS after serious improvement of its English and minor revisions.

The following inquiries should be addressed by the authors as minor revisions:

1- Page 1 L11: What do you exactly mean by "non-coherent granular flow equation"? May be it is better to mention your governing equation as "Shallow water type", "Depth-averaged type", "Incompressible Euler equations" or .... If the following explanations are the meaning of "non-coherent granular flow equation" write "non-coherent granular flow equation; i.e. Mih equation …" instead of "non-coherent granular flow equation. In this model, Mih equation …"

2- Page 1 L29: What is "landslide dam"? Is it suitable to be applied as a keyword for this paper?

3- Page 2 L15-16: "mesh-less methods" and "particle-based methods" can be considered as a one category as "mesh-less methods". There are two major categories: Mesh-based and Mesh-less methods.

4- Page 4 L8: The continuous granular flow model may consider the granular material as a viscous or inviscid fluid not necessarily a viscous fluid! Some rheological models such as Coulomb and Voellmy consider no viscosity or shear rate in their rheological formulations. Iverson (1997) has mentioned this rhelologies.

5- Page 4 L30: Parameter "*d*" in defined twice! One of them should be "*D*" which is missing here! Please correct the definition of the parameters of Eq. (1). Define parameters "*u*" and "*y*".

6- Page 5 L7: What do you mean by "dispersed pahse"?

7- Page 5 L24: Mention a reference for RNG k-ε model.

8- Page 13 Fig. 10: Are the locations of V0-V3 fixed or they move with the motion of landslide? Can you add a figure showing the average depth of landslide at different times in these four locations? Commonly, landslide velocities are sketched for landslide front, middle, and rear edges which move with the motion of the sliding mass along its path (For example see Yavari-Ramshe et al. 2015 in Computers and Geotechnics).

9- Page 16 L23: Please show the location of run-up data on a figure. For example, you can mark each point in Fig. 8 and use your marks in this table to show the location of each point.

10- The manuscript needs to be checked carefully regarding its English. A native English speaker should read through the entire text. Some of the cases are mentioned in the following.

- Page 1 L1: The title of the manuscript may be changed to "Analysis of the Tangjiaxi Landslide-generated waves in Zhexi Reservoir, China, by a granular flow coupling model".

- Page 1 L8: Write "A rocky granular flow commonly is formed after the failure of rocky bank slopes. An impulse wave disaster may also be initiated if the rocky granular flow rushes into a river with a high velocity." Instead of "Rocky granular flow usually forms after rocky bank slopes are failed and rushes into rivers at a high velocity, causing impulse wave disasters."

- Page 1 L10: Write "In this paper, …" instead of "In the paper, …".

- Page 1 L11: It is better to write "is developed based on …" instead of "is built based on …".

- Page 1 L13: Write "controls movements of the sliding mass" instead of "controls the movement of sliding mass". Please check the entire manuscript for the application of "the" wherever is needed.

- Page 1 L14: Write "water, and the Re-Normalisation Group (RNG) …" instead of "water, Re-Normalisation Group (RNG) …".

- Page 1 L15: Write "The proposed model is validated and applied for the 2014 Tangjiaxi landslide of Zhexi Reservoir located in Hunan Province, China, to analyze the characteristics of both landslide motion and its following impulse waves." instead of "Taking Tangjiaxi landslide as an example, which is located at Zhexi Reservoir in Hunan Province, China, the motion characteristics of Tangjiaxi landslide and the following impulse wave process were analyzed by the coupling model, and the validity of this model was checked."

- Page 1 L18: Write "On July 16, 2014, a rocky debris flow was formed after the failure of Tangjiaxi landslide, damming Tangjiaxi stream and causing an impulse wave disaster with three dead and nine missing bodies." instead of "On July 16, 2014, rocky blocks debris flow was formed after the failure of Tangjiaxi landslide, damming Tangjiaxi stream and thus causing an impulse wave disaster with which left three persons dead and nine persons missing."

- Page 1 L20: Write "Based on the full coupling numerical analysis, the granular flow impacts the water with a maximum velocity of about 22.5 m/s. Moreover, the propagation velocity of the

generated waves reach up to 12 m/s." instead of "The full coupling numerical analysis showed that after the failure of Tangjiaxi rockslide, rocky granular flows impacted the water at the maximum velocity of about 22.5 m/s, with waves propagating at the maximum celerity of up to 12 m/s."

- Page 1 L23-27: Write "The maximum calculated run-up of 21.8 m is close enough to the real value of 22.7 m. The predicted landslide final deposit and wave run-up heights are in a good agreement with the field survey data. These facts verify the ability of the proposed model for simulating the real impulse wave generated by rocky granular flow events" instead of "The deposited topographic modeled is similar to that accumulated in the actual situation. The maximum run-up calculated is 21.8 m, close to the value of 22.7 m obtained in the field survey. A series of run-up values in the field survey matches well with the calculated values. Therefore, the full coupling numerical model built in this study can be used to simulate impulse waves generated by rocky granular flows."

- Page 1 L32: Put "," after lakes as "rivers, lakes, and seas …".

- Page 1 L37: "formulae" is not an applicable word! Consider writing "A large number of researches have been done on landslide-induced impulse wave including analytical, physical, and numerical methods." instead of "A large number of researches have been done on landslide-induced impulse wave with formulae, physical experiment method and numerical analysis method.".

- Page 1 L38: Write "The analytical solutions are derived from …." Instead of "The formulae derive …".

- Page 1 L39: Write ", where their application scope is limited to their sources" instead of ", with its application scope closely related to sources".

- Page 1 L41: Write "Due to the considered simplifications for analytical solutions, …" instead of "Due to relatively simple results after calculation by the formulae, …".

- Page 2 L2: Write "the generation process of landslide impulse waves" instead of "the process of how landslide induces impulse waves".

- Page 2 L3-4: Write ". However, it requires a large amount of data, time, and money, and occupies a big space (Huang et al.2014)." instead of ", but it need large data, occupy big space, spend much money, and take a long time (Huang et al.2014).".

- Page 2 L6-7: Correct "it has the advantage of precise, economic and reasonable, as well as highly visible results (Heller et al. 2009)." As "it has the advantages of being precise, economic and reasonable, as well as having highly visible results (Heller et al. 2009).".

- Page 2 L7: Write "is an efficient tool" instead of "is an important tool".

- Page 2 L8: Correct "Yuvari-Ramshe" as "Yavari-Ramshe" within the entire manuscript.

- Page 2 L9: Consider "Regarding the granular mass/water body coupling system, three major numerical simulation methods have been recently applied, i.e." instead of "In the field of granular mass/water body coupled numerical analysis, three main numerical simulation methods are now used to analyze the landslide-induced impulse wave disaster, i.e.".

- Page 2 L10-12: Eliminate repeated "for landslide-induced impulse wave" as " a) single model, b) simplified model, and c)full coupling model …".

- Page 2 L13-17: Write "Each model may apply a mesh-based (e.g. finite difference method (FDM), finite element method (FEM), finite volume method (FVM), boundary element methods (BEM), et al.), or a particle-based (smoothed particle hydrodynamic (SPH), material particle method (MPM), et al.) method (Yavari-Ramshe and Ataie-Ashtiani, 2016) for numerical discretization of its model equations." Instead of "Their numerical calculation is constructed by the mesh-based methods (finite difference method (FDM), finite element method (FEM), finite volume method (FVM), boundary element methods (BEM), et al.), meshless-based methods (smoothed particle hydrodynamic (SPH), material particle method (MPM), et al.), and particle-based discrete element method (Yuvari-Ramshe and Ataie-Ashtiani, 2016).".

- Page 2 L22: Put "," after "Then" as "Then, various …"

- Page 2 L23: Write "to calculate the characteristics of the initial impulse wave …" instead of "to calculate initial impulse wave …".

- Page 2 L27-29: Write "Some examples of these models are TUNAMI" instead of "This type of numerical simulation models includes TUNAMI". And, mention the reference of each model in front of it (e.g. TUNAMI (), MOST (), ….).

- Page 2 L32: Some words such as "come up with" are not suitable academic words for applying in an academic writing. Please consider this point for the entire text. You may use "introduce" or "summarize" instead of "come up with".

- Page 2 L37: "and coupling calculated the impulse waves" vague statement. Please rewrite!

- Page 2 L38-39: Correct "at western Norway Åkerneset fjord" as "at Åkerneset fjord, western Norway".

- Page 3 L6-8: "The full coupling model for landslide-induced impulse wave, is a currently emerging method, which is booming recently, can have a relatively accurate description of the motion of sliding mass, interaction with water, and consequent generation, propagation and run-up of impulse waves" this is a very long sentence. Please rewrite this as two or three sentences.

- Page 3 L8-11: Some parts of the manuscript have grammatical mistakes. Please check the entire manuscript for English grammar. Correct "As a simple mathematical motion model has much difficulties in achieving real description of the motion of landslide, the model mostly used is the complicated rheological model or discrete element model." As "Simplified models have obvious difficulties in achieving an accurate description of the landslide motion. Accordingly, numerical models which consider the rheological behavior of the sliding mass in their calculations have been recently applied more often." Try to use simple short sentences which are more practical than using long complicated sentences!

- Page 3 L11-12: Write "The most applied continuous rheological models so far includes" instead of "In researches so far, models that describe flow-liked landslide or debris flow in continuous rheological models are".

- Page 3 L18: Correct "large deformation free surface" as "large free surface deformations".

- Page 3 L21: Write either "submarine landslide and tsunami. The landslide motion was …" or "submarine landslide and tsunami, where the landslide motion was …" instead of "submarine landslide and tsunami, the landslide motion was …".

- Page 3 L23: Correct "By combined landslide dynamic model and tsunami model, …" as "By combining a landslide dynamic model and a tsunami model, …".

- Page 3 L25: Finish the sentence after tsunami as "a landslide-induced tsunami. This model was applied to the 1792".

- Page 3 L27: Correct "In the paper, …" as "In this paper, …". Check thiswithin the entire manuscript.

- Page 3 L27-30: Write ""In this paper, a full coupling model is developed for landslide-induced impulse wave based on non-coherent granular flow equation. The continuous granular flow model of Mih (1999) is applied to simulate the motion process of the rocky granular flow after rockslide. Then, a two-phase flow model is adopted for granular mass / water interaction

coupled calculation." Instead of "In the paper, a full coupling model for landslide-induced impulse wave based on non-coherent granular flow equation is built and then the continuous granular flow Mih (1999) model is introduced to simulate the process of rocky granular motion after rockslide, and the two-phase flow model is adopted for interaction coupled calculation."

- Page 3 L38: Correct "After rocky slopes fail, high concentration and non-coherent rocky granular motion" as "The failure of a rocky slope is commonly followed by a high concentration and non-coherent rocky granular motion".

- Page 3 L43-44: "The discontinuous model features natural intuitive similarity when used to study the motion of non-coherent granular flows." Vague statement! Please rewrite!

- Page 4 L8: Mention "The present continuous granular flow model" instead of "The continuous granular flow model"

- Page 4 L9: Write "was studied by several researchers such as Bagnold (1954) …." Instead of "was studied by Bagnold (1954) ….".

- Page 4 L25-26: Write "He described the shear stress of a granular flow as follow:" instead of "The equation for shear stress of Mih (1999) granular flow is as follows:".

- Page 4 L28: Increase the writing quality of $\mu$ and $\rho$. Write "interestial fluid density" instead of "density between granular".

- Page 5 L24: Correct "when the granular flow into the water" as "when the granular flow enters the water".

- Page 6 L24-26: Rewrite "The case of Tangjiaxi landslide in Zhexi Reservoir, Hunan, China, is taken as an example, the whole process of the landslide and impulse wave induced are analyze, as well as the validity of numerical model." As "The Tangjiaxi landslide event in Zhexi Reservoir, Hunan, China, is simulated as an example to analyze the whole process of the landslide motion and the impulse wave generation, propagation, and runup."

- Page 6 L29-30" Write "destroyed the nearby residential area" instead of "destroyed resident living area nearby".

- Page 7 L4: Correct "much attention" as "more attention".

- Page 10 L16: Write "within its path" rather than "it met".

- Page 10 L18: Correct "six of which were badly hurt" as "six of them were badly hurt"

- Page 11 L3-4: Write "The computational domain which is considered to simulate the Tangjiaxi landslide-induced impulse wave by the full coupling numerical model covers the landforms of the valley where Tangjiaxi landslide occurred." Rather than "the full coupling numerical model for Tangjiaxi landslide-induced impulse wave is built based on the landforms of the valley where Tangjiaxi landslide occurred."

- Page 11 L4: Write "The domain is 792 m long and 684 m wide including the valley source of …" rather than "The model is 792 m long and 684 m wide. The model area covers the valley source of …"

- Page 11 L8-8: Eliminate "Tangjiaxi landslide model is set to be a granular flow model."

- Page 11 L11: Write "Thus, the sliding material is can be supposed to be saturated." Rather than "Thus, the fluid in Tangjiaxi landslide granular flow gaps was water."

- Page 11 L21: In Table 1, please mention the unit of each parameter.

- Page 12 L11: It is better to say "3.3 Numerical results" rather than "3.3 Results".

- Page 12 L12: Write "In this simulation, the following aspects of the Tangjiaxi landslide event are analyzed:" rather than "The coupled results were analyzed in the following aspects:"

- Page 12 L16: Write "The model analysis starts with the movement of the sliding mass" rather than "Upon the start of the model analysis, the sliding mass started to move.".

- Page 12 L16-17: Write "The depth-averaged velocity curves at different elevation points of the sliding mass show that the time of reaching to the maximum velocity is varied for different parts of the landslide" rather than "From the depth-averaged velocity curves at different elevation points in the sliding mass, it can be seen that the time that different parts of the sliding mass took to reach the maximum velocity varied."

- Page 12 L18-20: Write "Most of the landslide parts reached to the maximum velocity before impacting the opposite valley at the 6th second." instead of "Generally the parts of sliding mass reached the maximum velocity before the sliding mass impacted the opposite valley (the 6th second)."

- Page 13 L7: Correct "at different time" as "at different times".

- Page 13 L16-17: Write "The depth profile of Section A-A' (Fig. 12) in Fig. 13 shows that the solid grains of the sliding mass gradually moved toward the valley and accumulated." instead of

"From the A-A' section dynamic process of the landslide in Fig. 13, we can see that as the time went, solid grains of the sliding mass gradually moved to the valley and accumulated."

- Page 14 L4: Write "The slide front edge" rather than "the leading of the sliding mass".

- Page 14 L7: Write "and it remained unchanged forming a landslide dam …" rather than "and it almost kept unchanged from then on, thus forming a landslide dam …".

- Page 14 L14: Write "show no significant differences" rather than "don't show significant differences".

- Page 15 L10: Correct "impacted to houses in A" as "impacted the houses of area A".

- Page 15 L13-15: Write "Based on the numerical results, it has taken about 20 sec since the landslide start moving until the impulse waves reached the first residential area." rather than "Based on calculation, the duration from the time the sliding mass started to the time impulse waves attacked the houses was about 20 s."

- Page 15 L17-18: Write "As it can be seen in Fig. 2, the Tangjiaxi valley is narrow. Therefore, it is hard to distinguish the generation, propagation and run-up phases of the impulse wave. Accordingly, this event was not a typical landslide-induced impulse waves." rather than "We can also see from Fig. 2 that as Tangjiaxi valley was narrow, the phases of generation, propagation and run-up of the impulse wave were hard to distinguish at the reach where the landslide slid into water, so it was not a typical process of impulse waves."

- Page 16 L1-4: Write "As it can be observed in the water level lines of various points in Tangjiaxi river surface in Fig. 15, there was only one large peak for the impulse waves, close to the landslide impact area (H3 in Fig. 15)." rather than "As shown in the water level process line of various points in Tangjiaxi river surface (Fig. 15), there was only one large peak for the impulse waves in the landslide, especially typical at the reach where the landslide slid into water (H3 in Fig. 15)."

- Page 16 L18: Write "Table 2 shows …" rather than "Table 2 shew …".

- Page 16 L21: Write "are in a good agreement with" or "adequately match with" rather than "had high goodness of fit with".

- Page 16 L21: Please try to avoid long sentences within the entire manuscript. Write ". Thus, the numerical model is a valid and reasonable tool for simulating landslide-induced impulse wave hazards." Rather than ", so the numerical model for landslide-induced impulse wave is reasonable and valid."

- Page 16 L23: The title of Table 2 can be changed to "Table 2 The calculated and measured run-up values at different points".

-  Page 16 L26: Write "In this paper" rather than "In the paper" and "was developed" rather than "was built". These are repeated several times. Please check the entire manuscript. Also, finish the sentence after "was built," as "was developed.".

- Page 16 L28: Write "The non-coherent granular flow model of Mih (1999)" rather than "non-coherent granular flow Mih model". Correct "dynamic characteristic" as "dynamic characteristics".

- Page 17 L4: Eliminate one of the "wave" words.

---

## Author Response (AR1)

**Manuscript Number: NHESS-2016-332**
**Title: Tangjiaxi Landslide and Impulse Wave Analysis in Zhexi Reservoir of China by Granular Flow Coupling Model**

**Response letter**

Dear reviewers and editors:

We appreciated the thorough professional reviews provide by the journal and the positive comments from reviewers. Thank you for your detailed and good suggestion, I learn much from your comments. We do our best to modify the MS, and we think the MS is better than before. I have responded the comment point to point, you can see it as below.

Below is our response to the comments resulting in a number of clarifications.

Best wish to you!

Yours sincerely
Bolin Huang

**Reviewer1**

Response to reviewer 1#

The authors appreciate the positive comments from reviewer 1#, thank you for your good suggestions and comments, which make the MS better! Below is the response point to point:

1. Landslide moved velocity is very important to impulse wave, and the parameters for numerical simulation dominate the results right or not. What are the parameters for calculating the landslide moving velocity, and how to determine them? What about the parameters for fluid?
**Response**: The dynamic of landslide is very important to the formation of the impulse wave. The granular flow coupling model calculate the movement of landslide mainly by the equations of granular flow (Mih, 1999) and other general equations, such as mass continuity equation, momentum equation and energy equations, and so on. Therefore, the special control equation is the equations of granular flow, the parameters controlled the dynamic of landslide are grain density, grain diameter, and grain restitution coefficient, and so on, which can be seen in Table 1. The fluid is mainly controlled by viscosity of fluid, as it is water, a Newtonian fluid.

The parameters of density, average diameter and initial porosity of rock grains were determined through field survey and laboratory tests. The grain restitution coefficient in the first phase is obtained by back analysis or trial calculation, which is 0.2.

In the former MS, we do not tell how the restitution coefficient is determined, we added it in Page 11 L16-18, as following: "After trial calculation, 0.2 was taken in the first phase when the impact mainly occurred among grains, which makes the simulation results more realistic."

2. The run-up of the wave is shown in Fig.8. Please introduce the methods to obtain the heights of wave and the wave kind.
**Response:** It is easy to obtain these data in Euler method, as the code is Euler algorithm. We record the hydraulic data history of certain positons, for example the free water surface elevation, we can read the water elevation of every positons history along time went. Therefore, the height of wave, the wave kind or the run-up is obtained. In the former MS, we do not tell the code is Euler algorithm, we add this information in the MS in Line15of Page 12, as

following: "With the finite element/volume method with Euler algorithm adopted".

3. The landslide dam shape formed in the numerical simulation is different from the actual situation (Page 13, Line 12) in Fig. 12 and Fig.13. Is it the main reason that the spherical solid gains with similar grain size? How about the in fluence of soil or rock parameters? Or the parameters in the manuscript were not reasonable?

**Response**: Yes, the landslide dam shape formed in the numerical simulation is different from the actual situation in Fig. 12 and Fig.13. We think that the presumption of the spherical solid gains with similar grain size is the main influence factor. Let us suppose that some viscos fluid slide along a plane, they always deposit as a fan. As long as we adopt this presumption, the shape will not change fundamentally, regardless of the parameters of rock and soil. The parameters of rock may change the position of deposit, but the sharp will still like a fan. But if there are some impurity or grain in this fluid, the sharp of deposit may changes. I hope I have explain what we think about the different about the deposit sharps clearly.

4. The results in Table 2 have some differences between investigation and calculation. What are the reasons? Further discussions should be added to the results in the manuscript.

**Response**: Yes, the simulating values and investigation values are not exactly the same. Thanks for your suggestion, we add a paragraph to discuss the reason and show the possible solution in the future for this numerical method, which is as following: "The equations of Baglad and Mih were obtained from the experiments of sphere grains, and there is non-coherence among the grains. Although some parameters are taken by back analysis in the case, the dynamic capacity of sphere grains is bigger than grains with other sharp, which make the energy transferred to water higher. Meanwhile, in the actual situation, rock mass slides into water along with disintegrated. In the dynamic process, there should considerate general coherence to reflect these forces. Therefore, the run-up values simulated are larger than investigations in generally. Consideration of coherence and sharp of grain is a main modification direction for this granular flow coupling model, which might improve its realism for a wider range of applications."

5. There are some spell mistakes in the manuscript, i.e. in Fig.1 the Tangyangguang landslide or Tangyanguan landslide.

**Response**:Yes, it should be Tangyanguang. Thank you for your careful review, I modified the picture as following, and the MS will keep changes synchronized.

[Figure]

Meanwhile, there are also some mistakes in the former MS, I have modified it; for example, "asl." is not consistent in the former MS, we modified it.

**Reviewer2#**

The authors appreciate the positive comments from the reviewer 2#.

The following inquiries should be addressed by the authors as minor revisions:
1- Page 1 L11: What do you exactly mean by "non-coherent granular flow equation"? May be it is better to mention your governing equation as "Shallow water type", "Depth-averaged type", "Incompressible Euler equations" or …. If the following explanations are the meaning of "non-coherent granular flow equation" write "non-coherent granular flow equation; i.e. Mih equation …" instead of "non-coherent granular flow equation. In this model, Mih equation …"
Response: thanks for your careful review, I have modified the sentence as following: " non-coherent granular flow equation, i.e. Mih equation. ……"

2- Page 1 L29: What is "landslide dam"? Is it suitable to be applied as a keyword for this paper?
Response: Yes, it may be not suitable. I have changed it into "dynamic process", and dynamic process may be more suitable as the MS is to calculate the whole dynamic process of landslide-induced impulse wave.

3- Page 2 L15-16: "mesh-less methods" and "particle-based methods" can be considered as a one category as "mesh-less methods". There are two major categories: Mesh-based and Mesh-less methods.
Response: Yes, you are right. Basically, the particle-based method is mesh-less method. Thank you for your suggestion and remind. I have modified the sentence as following: " Each model may apply a mesh-based (e.g. finite difference method (FDM), finite element method (FEM), finite volume method (FVM), boundary element methods (BEM), et al.), or a particle-based (smoothed particle hydrodynamic (SPH), material particle method (MPM), et al.) method (Yavari-Ramshe and Ataie-Ashtiani, 2016) for numerical discretization of its model equations."

4- Page 4 L8: The continuous granular flow model may consider the granular material as a viscous or inviscid fluid not necessarily a viscous fluid! Some rheological models such as Coulomb and Voellmy consider no viscosity or shear rate in their rheological formulations. Iverson (1997) has mentioned this rhelologies.
Response: Thank you for your remind! I have modified the paragraph to express this meaning as following: "High concentration granular flow was studied by Bagnold (1954), Savage (1978), Hanes and Inman (1985), Wang and Campbell (1992), Iverson (1997) and Mih (1999). Some rheological models such as coulomb and Voellmy consider no viscosity or shear rate in their rheological formulations (Iverson, 1997). In this study, the continuous granular flow model is built by using viscous fluid."

5- Page 4 L30: Parameter "$d$" in defined twice! One of them should be "$D$" which is missing here! Please correct the definition of the parameters of Eq. (1). Define parameters "$u$" and "$y$".
Response: Sorry, the first "$d$" should be "$D$", I have modified. du/dy was explained in the former MS, the "u" and "y" are defined as following: "$u$ is the mean velocity of the granular flow, y is the distance along the direction vertical to the moving direction".

6- Page 5 L7: What do you mean by "dispersed pahse"?
Response: The dispersed phase is the material relative to the continuous material (or continuous phase). In this MS, the continuous phase is the water; the dispersed phase is the granular. So I changed the sentence as following "While for the dispersed phase or the granular…".

7- Page 5 L24: Mention a reference for RNG k-ε model.
Response: Yakhot and Orszag (1986) and Yakhot and Smith (1992) are given as the references for RNG k-ε model.

8- Page 13 Fig. 10: Are the locations of V0-V3 fixed or they move with the motion of landslide? Can you add a figure showing the average depth of landslide at different times in these four locations? Commonly, landslide velocities are sketched for landslide front, middle, and rear edges which move with the motion of the sliding mass along its path (For example see Yavari-Ramshe et al. 2015 in Computers and Geotechnics).

Response: Yes, commonly, the unfixed positions of velocities are recorded as the points moving. However, the locations of V0-V3 are fixed as the numerical method is constructed by Euler method. The author add one figure (Fig. 14) to show the average depth of landslide at different times in these four locations, and add Yavari-Ramshe et al. 2015 as a reference.

[Figure]

Fig. 14 Depth process plot of monitoring points in the sliding mass.

Page 16 L23: Please show the location of run-up data on a figure. For example, you can mark each point in Fig. 8 and use your marks in this table to show the location of each point.

Response: Thank you for your good suggestion. I have modified figure 8 and table 2.

[Figure]

Fig. 8 The plot of run-up of the impulse wave generated by Tangjiaxi landslide, and the photos describe the scene of houses and trees damaged where marked by A, B, C and D in the upper map.

Table 2 The calculated and measured run-up values at different points

| North Run-up (m) | Position | g | f | e | d | c | b | a |
|---|---|---|---|---|---|---|---|---|
| | Investigation | 2.4 | 3.7 | 5.9 | 7.3 | 22.7 | 19.5 | 11.8 |
| | Calculation | 3.3 | 3.6 | 6.5 | 7.0 | 21.8 | 17.3 | 12.1 |

| South Run-up (m) | Position | l | k | j | i | | | |
|---|---|---|---|---|---|---|---|---|
| | Investigation | 2.2 | 3.4 | 9.0 | 3.0 | | | |
| | Calculation | 3.2 | 4.1 | 9.2 | 3.7 | | | |

10- The manuscript needs to be checked carefully regarding its English. A native English speaker should read through the entire text. Some of the cases are mentioned in the following.

Page 1 L1: The title of the manuscript may be changed to "Analysis of the Tangjiaxi Landslide-generated waves in Zhexi Reservoir, China, by a granular flow coupling model".
Response: I have changed the title to "Analysis of the Tangjiaxi Landslide-generated waves in Zhexi Reservoir, China, by a granular flow coupling model".

Page 1 L8: Write "A rocky granular flow commonly is formed after the failure of rocky bank slopes. An impulse wave disaster may also be initiated if the rocky granular flow rushes into a river with a high velocity." Instead of "Rocky granular flow usually forms after rocky bank slopes are failed and rushes into rivers at a high velocity, causing impulse wave disasters."
Response: I have changed the first sentence of the abstract to "A rocky granular flow commonly is formed after the failure of rocky bank slopes. An impulse wave disaster may also be initiated if the rocky granular flow rushes into a river with a high velocity".

Page 1 L10: Write "In this paper, …" instead of "In the paper, …".
Response: I have replaced "the" with "this".

Page 1 L11: It is better to write "is developed based on …" instead of "is built based on …".
Response: I have replaced "built" with "developed".

Page 1 L13: Write "controls movements of the sliding mass" instead of "controls the movement of sliding mass". Please check the entire manuscript for the application of "the" wherever is needed.
Response: I have deleted "the" in this sentence.

Page 1 L14: Write "water, and the Re-Normalisation Group (RNG) …" instead of "water, Re-Normalisation Group (RNG) …".
Response: I have added "and the" in this sentence.

Page 1 L15: Write "The proposed model is validated and applied for the 2014 Tangjiaxi landslide of Zhexi Reservoir located in Hunan Province, China, to analyze the characteristics of both landslide motion and its following impulse waves." instead of "Taking Tangjiaxi landslide as an example, which is located at Zhexi Reservoir in Hunan Province, China, the motion characteristics of Tangjiaxi landslide and the following impulse wave process were analyzed by the coupling model, and the validity of this model was checked."
Response: Thank you for your careful work. I have changed the sentence as your writing.

Page 1 L18: Write "On July 16, 2014, a rocky debris flow was formed after the failure of Tangjiaxi landslide, damming Tangjiaxi stream and causing an impulse wave disaster with three dead and nine missing bodies." instead of "On July 16, 2014, rocky blocks debris flow was formed after the failure of Tangjiaxi landslide, damming Tangjiaxi stream and thus causing an impulse wave disaster with which left three persons dead and nine persons missing."
Response: Thank you for your careful work. I have changed the sentence as your writing.

Page 1 L20: Write "Based on the full coupling numerical analysis, the granular flow impacts the water with a maximum velocity of about 22.5 m/s. Moreover, the propagation velocity of the generated waves reach up to 12 m/s." instead of "The full coupling numerical analysis showed that after the failure of Tangjiaxi rockslide, rocky granular flows impacted the water at the maximum velocity of about 22.5 m/s, with waves propagating

at the maximum celerity of up to 12 m/s."
Response: I have change the sentence as following:" Based on the full coupling numerical analysis, the granular flow impacts the water with a maximum velocity of about 22.5 m/s. Moreover, the propagation velocity of the generated waves reaches up to 12 m/s."

Page 1 L23-27: Write "The maximum calculated run-up of 21.8 m is close enough to the real value of 22.7 m. The predicted landslide final deposit and wave run-up heights are in a good agreement with the field survey data. These facts verify the ability of the proposed model for simulating the real impulse wave generated by rocky granular flow events" instead of "The deposited topographic modeled is similar to that accumulated in the actual situation. The maximum run-up calculated is 21.8 m, close to the value of 22.7 m obtained in the field survey. A series of run-up values in the field survey matches well with the calculated values. Therefore, the full coupling numerical model built in this study can be used to simulate impulse waves generated by rocky granular flows."
Response:
Response: Thank you for your careful work. I have changed the sentence as your writing.

Page 1 L32: Put "," after lakes as "rivers, lakes, and seas …".
Response: I have added a "," after "lakes".

Page 1 L37: "formulae" is not an applicable word! Consider writing "A large number of researches have been done on landslide-induced impulse wave including analytical, physical, and numerical methods." instead of "A large number of researches have been done on landslide-induced impulse wave with formulae, physical experiment method and numerical analysis method.".
Response: I have changed the sentence as your suggestion.

Page 1 L38: Write "The analytical solutions are derived from …." Instead of "The formulae derive …".
Response: I have modified the sentence into "The analytical solutions are derived from extensive sources".

Page 1 L39: Write ", where their application scope is limited to their sources" instead of ", with its application scope closely related to sources".
Response: I have changed the sentence as your suggestion.

Page 1 L41: Write "Due to the considered simplifications for analytical solutions, …" instead of "Due to relatively simple results after calculation by the formulae, …".
Response: I have changed the sentence as your suggestion.

Page 2 L2: Write "the generation process of landslide impulse waves" instead of "the process of how landslide induces impulse waves".
Response: I have changed the sentence as following: "the dynamic process of landslide-induced impulse waves".

Page 2 L3-4: Write ". However, it requires a large amount of data, time, and money, and occupies a big space (Huang et al.2014)." instead of ", but it need large data, occupy big space, spend much money, and take a long time (Huang et al.2014).".
Response: I have changed the sentence as your suggestion.

Page 2 L6-7: Correct "it has the advantage of precise, economic and reasonable, as well as highly visible results (Heller et al. 2009)." As "it has the advantages of being precise, economic and reasonable, as well as having highly visible results (Heller et al. 2009).".
Response: I have modified the sentence into "Response: "it has the advantages of being precise, economic and reasonable, as well as having highly visible results (Heller et al. 2009)".

Page 2 L7: Write "is an efficient tool" instead of "is an important tool".
Response: I have replaced "important" with "efficient".

Page 2 L8: Correct "Yuvari-Ramshe" as "Yavari-Ramshe" within the entire manuscript.
Response: Sorry to make this mistake, I have corrected the names within the entire manuscript.

Page 2 L9: Consider "Regarding the granular mass/water body coupling system, three major numerical simulation methods have been recently applied, i.e." instead of "In the field of granular mass/water body coupled numerical analysis, three main numerical simulation methods are now used to analyze the landslide-induced impulse wave disaster, i.e.".
Response: I have modified the sentence into: "Regarding the granular mass/water body coupling system, three major numerical simulation methods have been recently applied, such as a) single model…"

Page 2 L10-12: Eliminate repeated "for landslide-induced impulse wave" as " a) single model, b) simplified model, and c)full coupling model …".
Response: I have deleted the repeat words "for landslide-induced impulse wave".

Page 2 L13-17: Write "Each model may apply a mesh-based (e.g. finite difference method (FDM), finite element method (FEM), finite volume method (FVM), boundary element methods (BEM), et al.), or a particle-based (smoothed particle hydrodynamic (SPH), material particle method (MPM), et al.) method (Yavari-Ramshe and Ataie-Ashtiani, 2016) for numerical discretization of its model equations." Instead of "Their numerical calculation is constructed by the mesh-based methods (finite difference method (FDM), finite element method (FEM), finite volume method (FVM), boundary element methods (BEM), et al.), meshless-based methods (smoothed particle hydrodynamic (SPH), material particle method (MPM), et al.), and particle-based discrete element method (Yuvari-Ramshe and Ataie-Ashtiani, 2016).".
Response: I have modified the sentences as your suggestion.

Page 2 L22: Put "," after "Then" as "Then, various …"
Response: I have added "," after "Then".

Page 2 L23: Write "to calculate the characteristics of the initial impulse wave …" instead of "to calculate initial impulse wave …".
Response: I have added the words of "the characteristics of the" in the sentence.

Page 2 L27-29: Write "Some examples of these models are TUNAMI" instead of "This type of numerical simulation models includes TUNAMI". And, mention the reference of each model in front of it (e.g. TUNAMI (), MOST (), ….).
Response: I have modified the sentence into "Some examples of these models are TUNAMI (Fumihiko et al. 2006), MOST (Titov and Gonzalez 1997), FUNWAVE (Joseph et al. 2003; Tappin et al. 2008), CLAWPACK (Randall 2006), etc."

Page 2 L32: Some words such as "come up with" are not suitable academic words for applying in an academic writing. Please consider this point for the entire text. You may use "introduce" or "summarize" instead of "come up with".
Response: I have replace "come up with" with "introduced". Meanwhile, I checked the MS for this point.

Page 2 L37: "and coupling calculated the impulse waves" vague statement. Please rewrite!
Response: I have deleted the word of "coupling" to make it clear.

Page 2 L38-39: Correct "at western Norway Åkerneset fjord" as "at Åkerneset fjord, western Norway".
Response: I have modified it as "at Åkerneset fjord, western Norway"

Page 3 L6-8: "The full coupling model for landslide-induced impulse wave, is a currently emerging method, which is booming recently, can have a relatively accurate description of the motion of sliding mass, interaction with water, and consequent generation, propagation and run-up of impulse waves" this is a very long sentence. Please rewrite this as two or three sentences.
Response: I have rewrite this sentence as following: "The full coupling model for landslide-induced impulse wave is a currently emerging method, which is booming recently. The full coupling model can have a relatively accurate description of the motion of sliding mass, interaction with water, and consequent impulse waves".

Page 3 L8-11: Some parts of the manuscript have grammatical mistakes. Please check the entire manuscript for English grammar. Correct "As a simple mathematical motion model has much difficulties in achieving real description of the motion of landslide, the model mostly used is the complicated rheological model or discrete element model." As "Simplified models have obvious difficulties in achieving an accurate description of the landslide motion. Accordingly, numerical models which consider the rheological behavior of the sliding mass in their calculations have been recently applied more often." Try to use simple short sentences which are more practical than using long complicated sentences!
Response: Thank you for your good suggestion. As a non-native English, we often cannot deal with the long complex sentences well. I will try to use short sentence to make the meaning clear. Thanks. This sentence was modified as your suggestion.

Page 3 L11-12: Write "The most applied continuous rheological models so far includes" instead of "In researches so far, models that describe flow-liked landslide or debris flow in continuous rheological models are".
Response: I have modified the sentence as your suggestion.

Page 3 L18: Correct "large deformation free surface" as "large free surface deformations".
Response: I have corrected it as "large free surface deformations".

Page 3 L21: Write either "submarine landslide and tsunami. The landslide motion was …" or "submarine landslide and tsunami, where the landslide motion was …" instead of "submarine landslide and tsunami, the landslide motion was …".
Response: I have modified the sentence as following: "submarine landslide and tsunami, where the landslide motion was…".

Page 3 L23: Correct "By combined landslide dynamic model and tsunami model, …" as "By combining a landslide dynamic model and a tsunami model, …".
Response: I have added two "a" into the sentence as your suggestion.

Page 3 L25: Finish the sentence after tsunami as "a landslide-induced tsunami. This model was applied to the 1792".
Response: As you suggested, I divided this sentence into two sentences.

Page 3 L27: Correct "In the paper, …" as "In this paper, …". Check thiswithin the entire manuscript.
Response: I have modified it into "in this paper" in this sentence, as well as the beginning sentence of the section of conclusion.

Page 3 L27-30: Write ""In this paper, a full coupling model is developed for landslide-induced impulse wave based on non-coherent granular flow equation. The continuous granular flow model of Mih (1999) is applied to simulate the motion process of the rocky granular flow after rockslide. Then, a two-phase flow model is

adopted for granular mass / water interaction coupled calculation." Instead of "In the paper, a full coupling model for landslide-induced impulse wave based on non-coherent granular flow equation is built and then the continuous granular flow Mih (1999) model is introduced to simulate the process of rocky granular motion after rockslide, and the two-phase flow model is adopted for interaction coupled calculation."
Response:
Response: As you suggested, I modified these sentences as following: "In this paper, a full coupling model is developed for landslide-induced impulse wave based on non-coherent granular flow equation. The continuous granular flow model of Mih (1999) is applied to simulate the motion process of the rocky granular flow after rockslide. Then, a two-phase flow model is adopted for granular mass / water interaction coupled calculation".

Page 3 L38: Correct "After rocky slopes fail, high concentration and non-coherent rocky granular motion" as "The failure of a rocky slope is commonly followed by a high concentration and non-coherent rocky granular motion".
Response: I have modified the sentence as your suggestion.

Page 3 L43-44: "The discontinuous model features natural intuitive similarity when used to study the motion of non-coherent granular flows." Vague statement! Please rewrite!
Response: I have modified the sentence as following:" The discontinuous model for particle flow simulation has a natural similarity."

Page 4 L8: Mention "The present continuous granular flow model" instead of "The continuous granular flow model"
Response: I have added the word of "present" in the sentence.

Page 4 L9: Write "was studied by several researchers such as Bagnold (1954) …." Instead of "was studied by Bagnold (1954) ….".
Response: The entire paragraph was modified into: "High concentration granular flow was studied by several researchers such as Bagnold (1954), Savage (1978), Hanes and Inman (1985), Wang and Campbell (1992), Iverson (1997) and Mih (1999). Some rheological models such as coulomb and Voellmy consider no viscosity or shear rate in their rheological formulations (Iverson, 1997). In this study, the present continuous granular flow model is built by using viscous fluid".

Page 4 L25-26: Write "He described the shear stress of a granular flow as follow:" instead of "The equation for shear stress of Mih (1999) granular flow is as follows:".
Response: Yes, I changed the sentence as your suggestion.

Page 4 L28: Increase the writing quality of and . Write "interestial fluid density" instead of "density between granular".
Response: I have added "fluid" in front of "density".

Page 5 L24: Correct "when the granular flow into the water" as "when the granular flow enters the water".
Response: I have corrected "into " as "enters".

Page 6 L24-26: Rewrite "The case of Tangjiaxi landslide in Zhexi Reservoir, Hunan, China, is taken as an example, the whole process of the landslide and impulse wave induced are analyze, as well as the validity of numerical model." As "The Tangjiaxi landslide event in Zhexi Reservoir, Hunan, China, is simulated as an example to analyze the whole process of the landslide motion and the impulse wave generation, propagation, and runup."
Response: I have corrected the sentence as "The Tangjiaxi landslide event in Zhexi Reservoir, Hunan, China, is simulated as an example to analyze the whole process of the landslide motion and the impulse wave".

Page 6 L29-30" Write "destroyed the nearby residential area" instead of "destroyed resident living area nearby".
Response: I have modified the sentence as your suggestion.

Page 7 L4: Correct "much attention" as "more attention".
Response: I have corrected "much" as "more".

Page 10 L16: Write "within its path" rather than "it met".
Response: I have corrected "it met" as "within its path".

Page 10 L18: Correct "six of which were badly hurt" as "six of them were badly hurt"
Response: I have corrected "which" as "them".

Page 11 L3-4: Write "The computational domain which is considered to simulate the Tangjiaxi landslide-induced impulse wave by the full coupling numerical model covers the landforms of the valley where Tangjiaxi landslide occurred." Rather than "the full coupling numerical model for Tangjiaxi landslide-induced impulse wave is built based on the landforms of the valley where Tangjiaxi landslide occurred."
Response: I have corrected as following: "The computational domain which is considered to simulate the Tangjiaxi landslide-induced impulse wave by the full coupling numerical model covers the landforms of the valley where Tangjiaxi landslide occurred".

Page 11 L4: Write "The domain is 792 m long and 684 m wide including the valley source of …" rather than "The model is 792 m long and 684 m wide. The model area covers the valley source of …"
Response: I have corrected it as "The domain is 792 m long and 684 m wide including the valley source of …".

Page 11 L8-8: Eliminate "Tangjiaxi landslide model is set to be a granular flow model."
Response: I have deleted the sentence.

Page 11 L11: Write "Thus, the sliding material is can be supposed to be saturated." Rather than "Thus, the fluid in Tangjiaxi landslide granular flow gaps was water."
Response: I have modified the sentence as "Thus, the sliding material is can be supposed to be saturated".

Page 11 L21: In Table 1, please mention the unit of each parameter.
Response: I have added the unit of each parameter.

Page 12 L11: It is better to say "3.3 Numerical results" rather than "3.3 Results".
Response: I have changed the title of 3.3 into "numerical results".

Page 12 L12: Write "In this simulation, the following aspects of the Tangjiaxi landslide event are analyzed:" rather than "The coupled results were analyzed in the following aspects:"
Response: I have changed the sentence into "In this simulation, the following aspects of the Tangjiaxi landslide event are analyzed:…".

Page 12 L16: Write "The model analysis starts with the movement of the sliding mass" rather than "Upon the start of the model analysis, the sliding mass started to move.".
Response: Thank you for your suggestion. I have modified the sentence into "The model analysis starts with the movement of the sliding mass".

Page 12 L16-17: Write "The depth-averaged velocity curves at different elevation points of the sliding mass show that the time of reaching to the maximum velocity is varied for different parts of the landslide" rather than "From the depth-averaged velocity curves at different elevation points in the sliding mass, it can be seen

that the time that different parts of the sliding mass took to reach the maximum velocity varied."
Response: I have modified the sentences as your suggestion.

Page 12 L18-20: Write "Most of the landslide parts reached to the maximum velocity before impacting the opposite valley at the 6th second." instead of "Generally the parts of sliding mass reached the maximum velocity before the sliding mass impacted the opposite valley (the 6th second)."
Response: I have changed the sentence as "Most of the landslide parts reached to the maximum velocity before impacting the opposite valley at the 6th second."

Page 13 L7: Correct "at different time" as "at different times".
Response: I have modified "time" into "times".

Page 13 L16-17: Write "The depth profile of Section A-A' (Fig. 12) in Fig. 13 shows that the solid grains of the sliding mass gradually moved toward the valley and accumulated." instead of   "From the A-A' section dynamic process of the landslide in Fig. 13, we can see that as the time went, solid grains of the sliding mass gradually moved to the valley and accumulated."
Response: I have replaced the former sentences into "The depth profile of Section A-A' (Fig. 12) in Fig. 13 shows that the solid grains of the sliding mass gradually moved toward the valley and accumulated".

Page 14 L4: Write "The slide front edge" rather than "the leading of the sliding mass".
Response: I have modified as your suggestion.

Page 14 L7: Write "and it remained unchanged forming a landslide dam …" rather than "and it almost kept unchanged from then on, thus forming a landslide dam …".
Response: I have changed the sentence as "and it remained unchanged forming a landslide dam…"

Page 14 L14: Write "show no significant differences" rather than "don't show significant differences".
Response: I have replaced the former words as "show no significant differences".

Page 15 L10: Correct "impacted to houses in A" as "impacted the houses of area A".
Response: I have correct it as "impacted the houses of area A".

Page 15 L13-15: Write "Based on the numerical results, it has taken about 20 sec since the landslide start moving until the impulse waves reached the first residential area." rather than "Based on calculation, the duration from the time the sliding mass started to the time impulse waves attacked the houses was about 20 s."
Response: I have rewritten the sentence as following: "Based on the numerical results, it has taken about 20 sec since the landslide start moving until the impulse waves reached the first residential area".

Page 15 L17-18: Write "As it can be seen in Fig. 2, the Tangjiaxi valley is narrow. Therefore, it is hard to distinguish the generation, propagation and run-up phases of the impulse wave. Accordingly, this event was not a typical landslide-induced impulse waves." rather than "We can also see from Fig. 2 that as Tangjiaxi valley was narrow, the phases of generation, propagation and run-up of the impulse wave were hard to distinguish at the reach where the landslide slid into water, so it was not a typical process of impulse waves."
Response: I have modified the sentences into "As it can be seen in Fig. 2, the Tangjiaxi valley is narrow. Therefore, it is hard to distinguish the generation, propagation and run-up phases of the impulse wave. Accordingly, this event was not a typical landslide-induced impulse waves".

Page 16 L1-4: Write "As it can be observed in the water level lines of various points in Tangjiaxi river surface in Fig. 15, there was only one large peak for the impulse waves, close to the landslide impact area (H3 in Fig. 15)." rather than "As shown in the water level process line of various points in Tangjiaxi river surface (Fig. 15),

there was only one large peak for the impulse waves in the landslide, especially typical at the reach where the landslide slid into water (H3 in Fig. 15)."
Response: I have rewritten the sentence as following: "As it can be observed in the water level lines of various points in Tangjiaxi river surface in Fig. 15, there was only one large peak for the impulse waves, close to the landslide impact area (H3 in Fig. 15)".

Page 16 L18: Write "Table 2 shows …" rather than "Table 2 shew …".
Response: I have corrected "shew" as "shows".

Page 16 L21: Write "are in a good agreement with" or "adequately match with" rather than "had high goodness of fit with".
Response: I have replaced it with "adequately match with".

Page 16 L21: Please try to avoid long sentences within the entire manuscript. Write ". Thus, the numerical model is a valid and reasonable tool for simulating landslide-induced impulse wave hazards." Rather than ", so the numerical model for landslide-induced impulse wave is reasonable and valid."
Response: Thank you for your patient and careful work, I checked and modified the long sentence in the MS. I rewrite it as "Thus, the numerical model is a valid and reasonable tool for simulating landslide-induced impulse wave hazards".

Page 16 L23: The title of Table 2 can be changed to "Table 2 The calculated and measured run-up values at different points".
Response: I have modified the title of Table 2 as your suggestion.

Page 16 L26: Write "In this paper" rather than "In the paper" and "was developed" rather than "was built". These are repeated several times. Please check the entire manuscript. Also, finish the sentence after "was built," as "was developed.".
Response: Yes, I have checked the entire manuscript, I have replaced "the" with "this", replaced "built"with "developed", I have checked it in the entire manuscript.

Page 16 L28: Write "The non-coherent granular flow model of Mih (1999)" rather than "non-coherent granular flow Mih model". Correct "dynamic characteristic" as "dynamic characteristics".
Response: I have replace the words with "The non-coherent granular flow model of Mih (1999)", and added "s" after "characteristic".

Page 17 L4: Eliminate one of the "wave" words.
Response: Yes, I have deleted the surplus "wave".

**End!**
**Thank you for your patience!**